# Large-sample hydrology – A few camels or a whole caravan?

Franziska Clerc-Schwarzenbach[1], Giovanni Selleri[2], Mattia Neri[2], Elena Toth[2], Ilja van Meerveld[1], Jan Seibert[1]

[1]Department of Geography, University of Zurich, Zurich, 8057, Switzerland

[2]Department of Civil, Chemical, Environmental, and Materials Engineering, University of Bologna, Bologna, 40136, Italy

*Correspondence to*: Franziska Clerc (franziska.clerc@geo.uzh.ch)

**Abstract.** Large-sample datasets containing hydrometeorological time series and catchment attributes for hundreds of catchments in a country, many of them known as "Camels" (catchment attributes and meteorology for large-sample studies), have revolutionized hydrological modelling and enabled comparative analyses. The Caravan dataset is a compilation of several ("Camels" and other) large-sample datasets with uniform attribute names and data structure. This simplifies large-sample hydrology across regions, continents, or the globe. However, the use of the Caravan dataset instead of the original Camels or other large-sample datasets may affect model results and the conclusions derived thereof. For the Caravan dataset, the meteorological forcing data are based on ERA5-Land reanalysis data. Here, we describe the differences between the original precipitation, temperature, and potential evapotranspiration ($E_{pot}$) data for 1252 catchments in the CAMELS-US, CAMELS-BR, and CAMELS-GB datasets and the forcing data for these catchments in the Caravan dataset. The $E_{pot}$ in the Caravan dataset is unrealistically high for many catchments but there are, not surprisingly, also considerable differences in the precipitation data. We show that the use of the forcing data from the Caravan dataset impairs hydrological model calibration for the vast majority of catchments, i.e., there is a drop in the calibration performance when using the forcing data from the Caravan dataset compared to the original Camels datasets. This drop is mainly due to the differences in the precipitation data. Therefore, we suggest extending the Caravan dataset with the forcing data included in the original Camels datasets wherever possible, so that users can choose which forcing data they want to use, or at least indicating clearly that the forcing data in Caravan come with a data quality loss and using the original datasets is recommended. Moreover, we suggest not using the $E_{pot}$ data (and derived catchment attributes, such as the aridity index) from the Caravan dataset and replacing these with (or based on) alternative $E_{pot}$ estimates.

## 1    Large-sample datasets as a game changer in hydrological modelling studies

Starting with the CAMELS (Catchment Attributes and MEteorology for Large-sample Studies) dataset for the US (Addor et al., 2017; Newman et al., 2015; in this paper referred to as CAMELS-US), large-sample datasets have been developed for several other countries (e.g., CAMELS-CL for Chile (Alvarez-Garreton et al., 2018), CAMELS-BR (Chagas et al., 2020a) and CABra (Almagro et al., 2021) for Brazil, CAMELS-GB for Great Britain (Coxon et al., 2020a), or CAMELS-CH for Switzerland (Höge et al., 2023)). We refer to these datasets with the time series of hydrometeorological measurements and information

on catchment attributes for hundreds of catchments as the Camels datasets. Because computational power had increased and cloud-computing had advanced when these datasets became available, hydrological models can now be run for hundreds of catchments in a reasonable timeframe. The Camels datasets offer new opportunities for catchment model- and comparison studies because they minimize the effort that is needed to compile and check hydrometeorological data from different datasets.

This is a great progress, not only for individual studies, but also for the comparability of different modelling approaches because comparisons are easier when different research groups use the same data for the same sets of catchments.

The Camels datasets have so far been used for different purposes. Examples are the exploration of the predictability of hydrologic signatures (Addor et al., 2018), the use thereof to cluster similar catchments and to explore their behaviour (Jehn et al., 2020), and to analyse the influence of catchment characteristics on runoff processes (Mathai and Mujumdar, 2022; McMillan

et al., 2022). The datasets have also been used to conceptualize models, e.g., to determine subsurface flow contributions to the hydrograph (Ranjram and Craig, 2022), to assess the value of limited or alternative data for regionalization (Pool et al., 2019, 2021) or hydrological model calibration (Meyer Oliveira et al., 2023), and to test the influence of changes in the meteorological forcing data on model performance (van Beusekom et al., 2022; Deng et al., 2024). They have, furthermore, been used to train long short-term memory models (Gauch et al., 2021; Kratzert et al., 2024; Lees et al., 2021).

The Caravan dataset (Kratzert et al., 2023a) goes further than the Camels datasets. As indicated by the name, referring to a group of camels, it is a compilation of (subsets of) large-sample datasets released earlier. When the Caravan dataset was released, it included the national datasets CAMELS-US (Addor et al., 2017b), CAMELS-BR (Chagas et al., 2020a), CAMELS-GB (Coxon et al., 2020a), CAMELS-CL (Alvarez-Garreton et al., 2018), and CAMELS-AUS (Fowler et al., 2021), the North American dataset HYSETS (Arsenault et al., 2020), and the Central European dataset LamaH-CE (Klingler et al., 2021). The

Caravan dataset not only combined parts of these existing datasets but also solved issues related to the lack of comparability among the different datasets, and the lack of an index referring to human impacts for some of the datasets (Addor et al., 2020). The use of the globally available ERA5-Land (European ReAnalysis) data (Muñoz-Sabater et al., 2021) for all catchments in the Caravan dataset furthermore allows the extension of the dataset with catchments for which streamflow but no meteorological data are available. With this possibility, Caravan allows catchments in underrepresented (climatic) regions to be included

in a well-known large-sample dataset. This is positive and may be a first step towards a more equal representation of different regions and biogeoclimatic zones in hydrological research. Because of the use of reanalysis data for the forcing data, it is easier to update the Caravan dataset with additional forcing data, or a new version thereof, than when station data are used. Another advantage of the Caravan dataset as a standard resource for catchment data is that some of the catchments added by the community are not available as individual Camels datasets, i.e., the attributes and hydrometeorological time series for these catch-

ments can only be accessed via the Caravan dataset. Thanks to the open code and software, the Caravan dataset can be extended by the community. The number of catchments in the Caravan dataset had already grown to almost 13,000 (not counting duplicates) in February 2024. Acquiring data from Caravan is the easiest way to get started for large-sample model studies. However, we argue that using the Caravan data set instead of the individual Camels datasets may have disadvantages despite the obvious advantage of the convenience of using one large dataset instead of the individual datasets.

Following the Caravan philosophy of using the same data source for all catchments for all climatic variables, the meteorological forcing data in the original Camels datasets were replaced by reanalysis data from ERA5-Land. ERA5-Land (Muñoz-Sabater et al., 2021) is a component of the Copernicus Climate Change Service (C3S). With ERA5-Land, global time series of the water and energy cycle over land are described with 50 different variables. Compared to the earlier products ERA5 (31 km; Hersbach et al., 2020) and ERA-Interim (80 km; Dee et al., 2011), the spatial resolution (9 km) and the representation of

the water cycle improved for ERA5-Land (Muñoz-Sabater et al., 2021). However, ERA5-Land tends to overestimate potential evapotranspiration ($E_{pot}$) considerably (Klingler et al., 2021; Xu et al., 2024). $E_{pot}$ is computed differently in ERA5-Land than in ERA5 (as per rectification in the ERA5-Land data documentation (2024) on 18 November 2021). In ERA5, vegetated land is set to "crops/mixed farming" and it is assumed that there is no soil moisture limitation for the computation of $E_{pot}$. In ERA5-Land, evaporation from an open water surface (i.e., pan evaporation) is computed. The atmosphere is assumed to be unaffected

by the evaporation for both ERA5 and ERA5-Land.

In this paper, we describe the differences between the meteorological forcing data for the catchments in three Camels datasets (CAMELS-US, CAMELS-BR, and CAMELS-GB) and the ERA5-Land data in the Caravan dataset. We, furthermore, assess the consequences of the substitution of largely station-based data in the Camels datasets by the reanalysis data in the Caravan dataset on the calibration performance of a bucket-type rainfall-runoff model. It is important to raise awareness of these dif-

ferences, and their consequences on model results because the well-organized data structure and ease of access make it very tempting to use the Caravan dataset instead of the original Camels datasets, especially when conducting studies across multiple countries or geographic regions.

## 2    Caravan forcing data based on ERA5-Land

In the original Camels (and other large-sample) datasets, the forcing data were selected with respect to the data availability for

the region of interest. They were mainly based on station data, but for some regions, they also included satellite data or reanalysis data (Table 1). In most cases, several forcing data time series were included to allow the user to choose the most suitable one or to allow a comparison between different data inputs. When a catchment is added to Caravan, all forcing data are replaced with data from the ERA5-Land reanalysis dataset (Muñoz-Sabater et al., 2021).

Several studies have assessed the ERA5-Land reanalysis data by comparing it to station data. ERA5-Land temperature and

precipitation data were found to better match the observations for flatter regions than for regions with complex terrain (Almeida and Coelho, 2023; Gomis-Cebolla et al., 2023; Tan et al., 2023). Temperature data from ERA5-Land were considered to be good for Portugal (Almeida and Coelho, 2023), northeastern Brazil (Araújo et al., 2022), the Chinese Qilian mountains (Zhao and He, 2022), and Italy (Vanella et al., 2022). For Turkey, ERA5-Land underestimated the daily temperature, but represented temperature trends well (Yilmaz, 2023). For the Kelantan basin in Malaysia, the daily maximum temperatures were

underestimated and the daily minimum temperatures were overestimated (Tan et al., 2023). In their evaluation of ERA5-Land

data for Italy, Vanella et al. (2022) found that the variables of ERA5-Land can be used to estimate evapotranspiration. Regarding precipitation, Gomis-Cebolla et al. (2023) found that ERA5-Land represented the spatial and temporal precipitation patterns well for Spain, but also that there were some difficulties in representing complex precipitation patterns. They furthermore found that ERA5-Land tended to overestimate light precipitation events and underestimated heavier precipitation. This was also observed for the Tibetan Plateau (Wu et al., 2023) and the Kelantan basin in Malaysia (Tan et al., 2023). For the Tibetan Plateau, the overestimation of light precipitation led to an overestimation of annual precipitation (Wu et al., 2023). ERA5-Land also overestimated precipitation for China (Xie et al., 2022), but there were regional differences. ERA5-Land represented precipitation for northeastern China better than for southwestern China (Xie et al., 2022).

A number of previous studies have analysed the advantages and disadvantages of the gridded products of the ERA family when used as forcing data in hydrological models. For example, Beck et al. (2017) included ERA-Interim data in a comparison of different precipitation products with gauge data. They found a reasonable agreement between the ERA-Interim data and the gauged data for all regions of the world, except northern South America, Africa, Central Asia and Southeast Asia. Essou et al. (2016, 2017) compared different reanalysis products (including ERA-Interim) for North America and found that the datasets had similar temperature data, but that there was a bias in precipitation for the humid continental and subtropical regions (i.e., for the eastern part of the US) and that this led to a deterioration in model performance (Essou et al., 2016). However, the reanalysis data performed better than gridded data for large and mountainous catchments, where the density of weather stations is low (Essou et al., 2017). Based on these findings, they suggested using reanalyses as meteorological forcing data when observational data are missing or limited. Similarly, Tarek et al. (2020) tested ERA5 temperature and precipitation data for hydrological modelling in North America. They found a clear improvement in model performance compared to ERA-Interim data, and that model performances were similar to those achieved with observational data, except for the eastern half of the US. They concluded that ERA5 data are useful, especially when observational data are lacking. Baez-Villanueva et al. (2021) compared ERA5 precipitation data and three other precipitation products for Chile and found a similar model performance for ERA5 data and some of the gauge-corrected precipitation products. However, they also reported some difficulties with ERA5 data for snow-dominated catchments.

## 3 Assessment of the differences between Camels and Caravan forcing data

### 3.1 Choice of catchments and climate variables

We compared the precipitation, temperature, and potential evapotranspiration ($E_{pot}$) data for 1252 catchments in the Caravan dataset with the original forcing data from the CAMELS-US, CAMELS-BR, and CAMELS-GB datasets. We chose precipitation, temperature, and $E_{pot}$ data for the comparisons because they are the most relevant for hydrological modelling. From the different Camels forcing datasets, we chose those with the highest spatial resolution (see Table 1), except for precipitation for the Brazilian catchments (as described below). The period for the comparisons ranged from April 1983 to March 2013 for the

Brazilian catchments (southern hemisphere) and from October 1983 to September 2013 for the catchments in the US and Great Britain (northern hemisphere) to account for the differences in the water year for the two hemispheres.

For each catchment we compared the mean annual precipitation, the mean daily temperature and the mean annual $E_{pot}$. We only compared the mean annual values, even though there are other components of the time series, such as the timing of the rainfall events, that are also crucial for hydrological modelling. To account for differences in the data other than the mean values, we used a hydrological modelling approach (see Sect. 4) that implicitly takes into account all the features of the forcing time series through the simulation of streamflow. For hydrological modelling, the temperature data "per se" (i.e., when not considered as the driver of $E_{pot}$) are mainly relevant for snow-related processes, i.e., to determine if precipitation is falling as snow (and is thus stored in the catchment) and if the precipitation that accumulated as snow is melting. Hence, the accuracy of the temperature data is relevant for only a few days per year for catchments where snow is an essential component of the water balance. In other words, the temperature plays a minor role for hydrological modelling compared to the accuracy of the precipitation or $E_{pot}$ data (cf. Tarek et al., 2020). Still, we compared the temperature data for all catchments and focused on the mean daily temperature (rather than, for example, the number of days with temperatures below or above 0 °C).

When we compared the two datasets, we always subtracted the value from the Camels dataset from the value from the Caravan dataset (i.e., a positive difference indicates a larger value for the Caravan dataset and a negative difference indicates a smaller value for the Caravan dataset). To determine the relative differences (i.e., for the mean annual precipitation and $E_{pot}$), we divided this difference by the value from the Camels dataset and report it as a percentage. As the catchment characteristics that depend on the meteorological data also differ for the Caravan and Camels datasets, we furthermore compared the differences in the aridity index ($E_{pot}/P$).

### 3.2    Choice of Camels forcing data

The Camels forcing data we used for comparison had a spatial resolution of 1 km for CAMELS-US and CAMELS-GB, and a coarser resolution for CAMELS-BR (Table 1). For the US catchments, we used the Daymet v2 data (Thornton et al., 2014; see also Thornton et al., 2021) for precipitation and temperature (the mean daily temperature was estimated from the average of the daily minimum and maximum temperature). As $E_{pot}$ data are not available in the CAMELS-US dataset, we calculated $E_{pot}$ with the Priestley-Taylor formula (Priestley and Taylor, 1972) based on the input data from Daymet v2. This is in line with the suggestion by Newman et al. (2015) and similar to the approach used in earlier studies with CAMELS-US data (e.g., Seibert and Vis (2016) and Addor et al. (2018)). As input data for the $E_{pot}$ calculations, we used the elevation and latitude of each catchment, and the time series of the day of the year, day length, minimum and maximum temperature, vapor pressure, and solar radiation. The Priestley-Taylor coefficient was set to 1.26 (cf. Priestley and Taylor, 1972) for all catchments. For the catchments in Brazil (BR), we used the MSWEP v2.2 precipitation data (Beck et al., 2019), the CPC temperature data (NOAA, 2019), and the GLEAM v3.3a $E_{pot}$ data (Martens et al., 2017; Miralles et al., 2011), which are based on the Priestley-Taylor formula with satellite-derived radiation and air temperature data. We chose MSWEP v2.2 data for the precipitation instead of CHIRPS (Funk et al., 2015) because the MSWEP v2.2 daily time series are based on a data point every three hours, and the

160 ones from CHIRPS are based on one data point every five days, disaggregated to daily values via reanalysis. For the catchments in Great Britain (GB), we used the CEH-GEAR precipitation data (Keller et al., 2015; Tanguy et al., 2016), the CHESS-met temperature data (Robinson et al., 2017a), and the CHESS-PE $E_{pot}$ data (Robinson et al., 2016, 2017b), which are based on the Penman-Monteith formula, with meteorological data obtained from stations.

**Table 1: Meteorological source datasets from the Camels datasets and the Caravan dataset used for comparison.**

| Region | Variable(s) | Dataset | Spatial resolution | References |
|---|---|---|---|---|
| US | Precipitation, Temperature | Daymet v2 (based on station data) | 1 km | Thornton et al. (2014, 2021) |
| BR | Precipitation | MSWEP v2.2 (based on station-, satellite-, and rea-nalysis data) | 0.1 ° | Beck et al. (2019) |
| BR | Temperature | CPC (based on station data) | 0.5 ° | NOAA (2019) |
| BR | $E_{pot}$ | GLEAM v3.3a (Priestley-Taylor method based on sat-ellite data) | 0.25 ° | Martens et al. (2017), Miralles et al. (2011) |
| GB | Precipitation | CEH-GEAR (based on station data) | 1 km | Keller et al. (2015), Tanguy et al. (2016) |
| GB | Temperature | CHESS-met (based on station data) | 1 km | Robinson et al. (2017a) |
| GB | $E_{pot}$ | CHESS-PE (Penman-Monteith based on CHESS-met data) | 1 km | Robinson et al. (2016, 2017b) |
| US, BR, GB | Precipitation, Temperature, $E_{pot}$ | ERA5-Land (Penman-Monteith based on reanalysis data) | 9 km | Muñoz-Sabater et al. (2021) |

## 3.3 Differences between ERA5-Land data in the Caravan dataset and forcing data in the Camels datasets

### 3.3.1 Differences in mean annual precipitation

The mean annual precipitation in the Caravan dataset differed between -53 % and 101 % from the one in the Camels dataset,
i.e., taking the Camels data as a reference, the mean annual precipitation was underestimated by up to 53 % and overestimated by up to 101 % in the Caravan dataset. For 583 of the 1252 catchments (47 %), the deviation was within ±10 %, and for 968 catchments (77 %), it was within ±20 % (Fig. 1). The mean annual precipitation in the Caravan dataset was lower than in the CAMELS-US dataset for the catchments in the eastern part of the US and on the West Coast. For some catchments in the centre of the US, the mean annual precipitation in the Caravan dataset was much higher (>40 %) than in the CAMELS-US
dataset. For the southern part of Brazil, the mean annual precipitation in the Caravan dataset was almost consistently higher

(and sometimes much higher) than in the CAMELS-BR data, while for the northern part of Brazil, it tended to be lower than in the CAMELS-BR dataset. For the catchments in the eastern part of Great Britain, the mean annual precipitation was slightly higher in the Caravan dataset than in the CAMELS-GB dataset, while for the catchments in the western part of Great Britain, the mean annual precipitation was lower in the Caravan dataset than in the CAMELS-GB dataset.

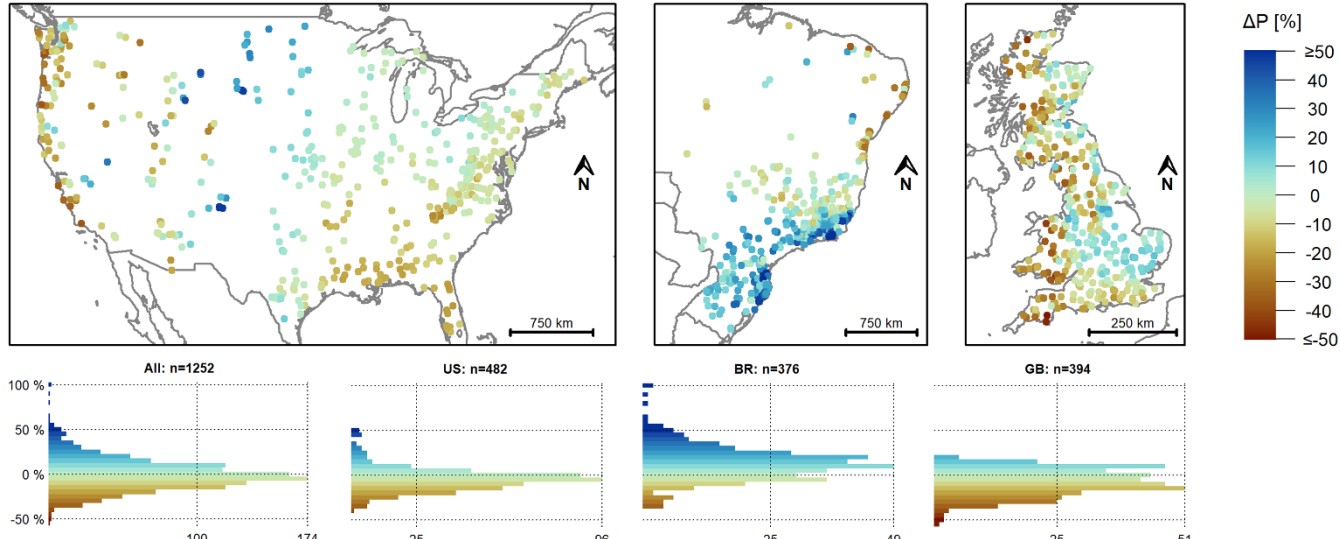

**Figure 1: Relative difference in the mean annual precipitation (calculated for a 30-year period: 1983-2013) for each catchment in the Caravan dataset compared to the mean annual precipitation for each catchment in the Camels datasets. The brown colours indicate less precipitation in the Caravan dataset, the blue colours indicate more precipitation in the Caravan dataset than in the Camels datasets. Note that the colour scale was cut at ±50 % but the histograms cover the full range of differences (at 5 % intervals). For one catchment, the difference was less than -50 % and for twelve catchments, it was more than 50 %. The scale bars refer to the map centre and are different for each country. The base maps with the country outlines were obtained from Natural Earth (naturalearthdata.com).**

### 3.3.2 Differences in mean daily temperature

The mean daily temperature data in the Caravan and Camels datasets were relatively similar. In the most extreme cases, the mean daily temperature in the Caravan dataset was 4 °C less (i.e., colder) and 2.8 °C higher (i.e., warmer) than in the Camels datasets. For 961 of the 1252 catchments (77 %), the temperature difference was less than ±1 °C (Fig. 2). For the catchments in the eastern part and the southern part of the West Coast of the US, the mean daily temperature in the Caravan dataset tended to be slightly higher than in the CAMELS-US dataset. For the catchments in the Pacific Northwest and most of the western US, the mean daily temperature in the Caravan dataset was lower than in the CAMELS-US dataset. In the snow-dominated Rocky Mountain region, the mean daily temperature in the Caravan dataset was up to 2.8 °C lower than in the CAMELS-US dataset. For Brazil, the mean daily temperature in the Caravan dataset was almost always lower than in the CAMELS-BR dataset (i.e., it was higher for only eight catchments), and this difference was often substantial. For 246 Brazilian catchments

(65 %), the mean temperature differed by at least -1 °C. For the catchments in Great Britain, the temperature data were similar, with differences between the two datasets varying between -0.9 °C and 0.5 °C.

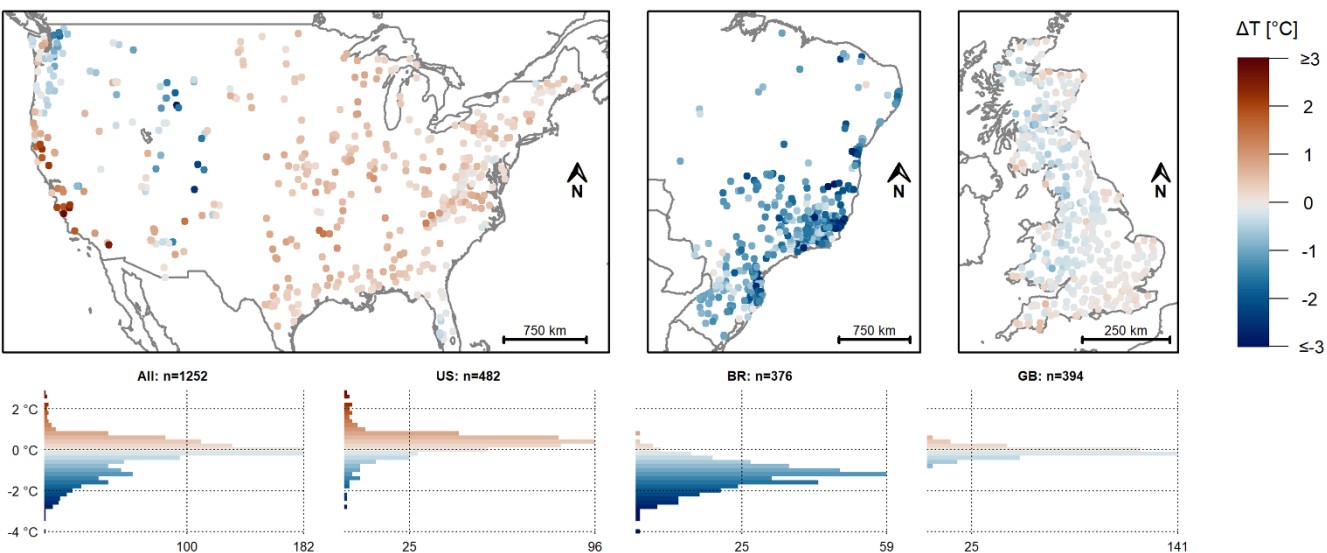

**Figure 2: Difference in the mean daily temperature (calculated for a 30-year period: 1983-2013) for each catchment in the Caravan**
**dataset and the Camels datasets. The blue colours indicate a lower mean daily temperature in the Caravan dataset, the red colours indicate a higher mean daily temperature in the Caravan dataset than in the Camels datasets. Note that the colour scale was cut at ±3 °C, but the histograms cover the full range of values (at 0.2 °C intervals). For three catchments, the difference was below -3 °C. For none of the catchments, the difference was higher than 3 °C.**

### 3.3.3     Differences in mean annual potential evapotranspiration

The $E_{pot}$ data derived from ERA5-Land in the Caravan dataset are unrealistically high for most catchments in the US, Brazil, and Great Britain (Fig. 3), confirming the results of Klingler et al. (2021) for Central Europe and Xu et al. (2024) for China. The minimum mean annual $E_{pot}$ in the Caravan dataset was higher than the maximum mean annual $E_{pot}$ in the Camels datasets for each of the three regions, i.e., the ranges of the $E_{pot}$ data did not overlap. The relative differences between the mean annual
$E_{pot}$ in the Caravan dataset and the mean annual $E_{pot}$ in the Camels datasets varied between 46 % and 913 % (median: 462 %) for the US catchments, between 58 % and 523 % (median: 121 %) for the Brazilian catchments, and between 52 % and 337 % (median: 120 %) for the catchments in Great Britain.

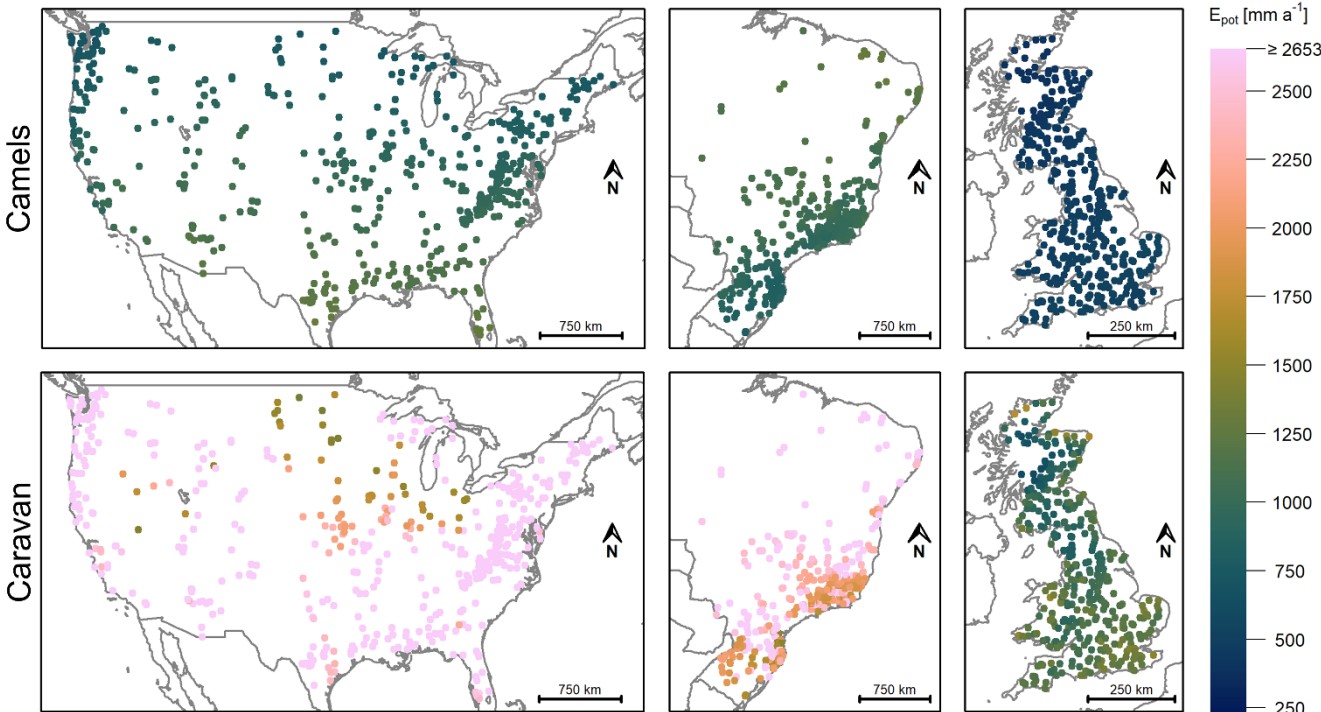

Figure 3: Mean annual $E_{pot}$ (calculated for a 30-year period: 1983-2013) for the Camels datasets (Brazil, Great Britain) or calculated with the data from the Camels dataset (US) (top) and for the Caravan dataset (bottom). Note that the colour scale ends at twice the maximum $E_{pot}$ value reported in the Camels datasets. The number of catchments for which the $E_{pot}$ in the Caravan dataset was higher than this cutoff value (2653 mm a$^{-1}$, shown in light pink) was 385 for the US (80 % of the US catchments), 115 for Brazil (31 %), and 0 for Great Britain.

Even though the use of $E_{pot}$ from the ERA5-Land data is consistent with the other variables in the Caravan dataset, the high (and often unrealistic) $E_{pot}$ values are problematic. Kratzert et al. (2023a) mention the high $E_{pot}$ values in the Caravan paper in a table caption. However, hydrologists using the Caravan dataset under the assumption that the data are ready for use may end up with wrong conclusions. The high $E_{pot}$ values do not only influence model simulation results (see Sect. 4.2) but also the catchment attributes based on these values. For the 30 years considered here, the mean annual $E_{pot}$ was larger than the mean annual precipitation (i.e., the aridity index was larger than 1.0) for 1059 of the 1252 catchments (85 %) based on the Caravan data whereas this was the case for only 167 catchments (13 %) based on the Camels data (Fig. 4).

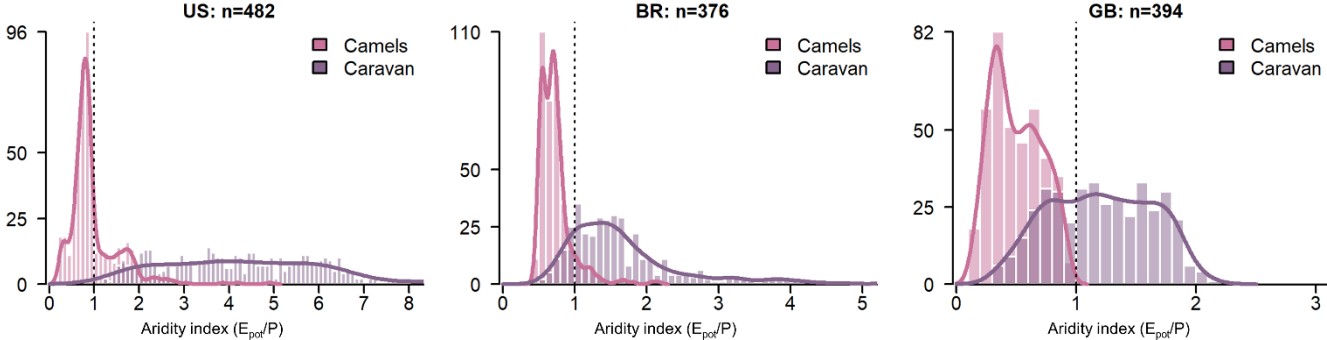

**Figure 4: Histograms of the aridity index values based on the mean annual evapotranspiration ($E_{pot}$) and precipitation (P) from the Camels and Caravan datasets (calculated for a 30-year period: 1983-2013). Note that 39 US catchments (8 %), and 4 Brazilian catchments (1 %) were not included in the histograms because the aridity index values for the Caravan data plot beyond the x-axis limits. The maximum calculated aridity index values were 20.2 for the US, 8.1 for Brazil, and 2.2 for Great Britain.**

To provide a possible alternative, we calculated time series of $E_{pot}$ using the formula given by Adam et al. (2006) based on Droogers and Allen (2002). This formula is based on the Hargreaves formula (Hargreaves and Samani, 1982) and was used for one of the $E_{pot}$ products included in the CAMELS-AUS dataset (Fowler et al., 2021). The relatively low data requirement for this method allowed us to calculate $E_{pot}$ time series based on the ERA5-Land precipitation and temperature data only, i.e., not violating the philosophy of Caravan to use only globally available data. More specifically, it takes only the location and temperature into account, and additionally adjusts the $E_{pot}$ estimates based on the monthly precipitation as a proxy for humidity. As input data, we used the latitude of each catchment, as well as the time series of the day of the year, daily mean temperature, the difference between the mean daily maximum temperature and the mean daily minimum temperature for each month, and the monthly precipitation sums (see the data repository linked in the data availability statement for the calculations). We refer to this $E_{pot}$ data as "Hargreaves $E_{pot}$".

The Hargreaves $E_{pot}$ data resulted in a mean annual $E_{pot}$ that was similar to the one of the Camels datasets (Table 1). For the US, the ratio between the mean annual Hargreaves $E_{pot}$ and the mean annual $E_{pot}$ in the Camels datasets varied between 0.6 and 1.4 (median: 0.9). This range was 0.6 to 1.3 for the catchments in Brazil (median: 1.0), and 0.5 to 1.1 for the catchments in Great Britain (median: 0.9). The catchments in the US and Great Britain for which the Hargreaves $E_{pot}$ values were (too) low were mainly located at the higher latitudes. As a comparison, the ratio between the mean annual $E_{pot}$ in the Caravan dataset and the mean annual $E_{pot}$ in the Camels datasets varied between 1.5 and 10.1 (median: 5.6) for the US, between 1.6 and 6.2 (median: 2.2) for Brazil, and between 1.5 and 4.4 (median: 2.2) for Great Britain (Fig. 5).

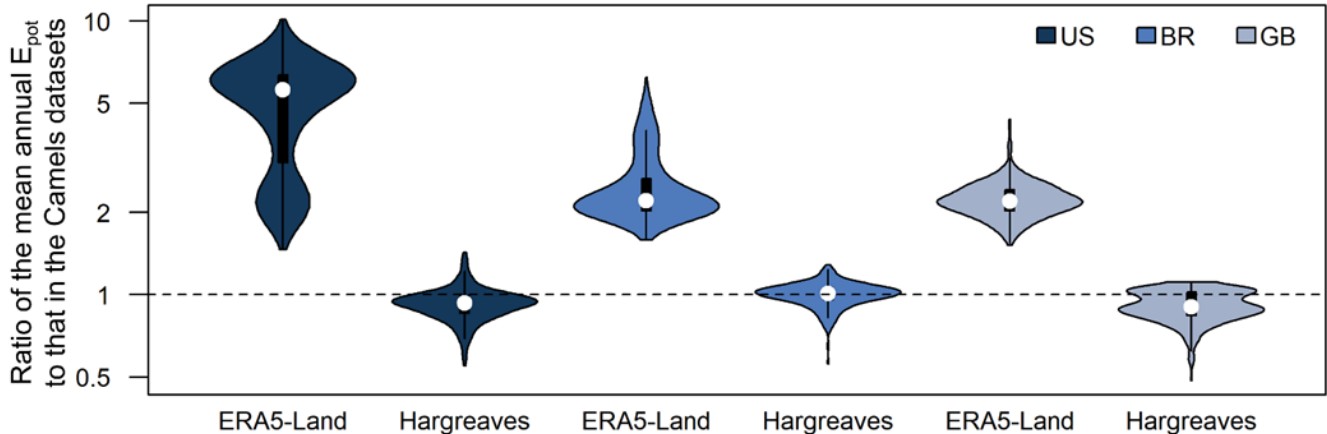

**Figure 5: Violin plots showing the ratio between the mean annual $E_{pot}$ of either the ERA5-Land data in the Caravan dataset or the Hargreaves $E_{pot}$ data based on input data from the Caravan dataset and the mean annual $E_{pot}$ in the Camels datasets for the catchments in the US, Brazil and Great Britain. The Camels $E_{pot}$ refer to the $E_{pot}$ data calculated with the Priestley-Taylor equation for the CAMELS-US dataset and the $E_{pot}$ data included in CAMELS-BR and CAMELS-GB datasets (see Table 1). Note that the y-axis is logarithmic.**

Of course, there are a variety of other ways to obtain daily $E_{pot}$ values for the Caravan dataset and to provide an alternative to the current $E_{pot}$ data in the Caravan dataset, e.g., the Hargreaves-Samani equation without the adjustment for humidity (Hargreaves and Samani, 1982) or the Thornthwaite equation (Thornthwaite, 1948) with a scaling for daily values. While it is an open question which method leads to the best results, the Hargreaves-based method used here provides a straightforward solution to avoid the problematic ERA5-Land-based Caravan $E_{pot}$ data.

## 4    Effect of the differences in the forcing data on hydrological model results

### 4.1    Description of modelling experiments

To assess the overall effect of the differences in the forcing data for the Camels and the Caravan datasets on hydrological model performance, we conducted a series of modelling experiments. Even though a compensational effect of the model parameters can be expected, i.e., to adjust for possibly inaccurate or biased forcing data, we consider the model performances (i.e., how well the streamflow observations could be represented with a certain combination of forcing data) as an aggregated measure for data quality.

We calibrated the bucket-type HBV model (Bergström, 1992; Lindström et al., 1997) in the version HBV-light (Seibert and Vis, 2012) with a genetic algorithm (Seibert, 2000), optimizing the Kling-Gupta efficiency (KGE; Gupta et al., 2009) for the daily streamflow simulations. A detailed description of the model routines can be found elsewhere (e.g., Seibert and Vis, 2012).

We created seven different combinations of forcing data, varying the data source for the precipitation, temperature, and $E_{pot}$ time series (Table 2), and calibrated the model for each of these datasets. We did this for each of the 1252 catchments for

which we also compared the forcing data (see Sect. 3.3). These are all catchments from CAMELS-US, CAMELS-BR, and CAMELS-GB that were included in the Caravan dataset, except for 14 catchments from CAMELS-GB for which more than 20 % of the streamflow data were missing for the simulation period. We divided each catchment into elevation zones of 200
285 m, whereby each elevation zone had to make up at least 5 % of the catchment area (if not, the elevation zones were merged with the neighbouring elevation zone). This division is relevant for the snow routine of the HBV model. We used the EarthEnv-DEM90 digital elevation model (Robinson et al., 2014) and the shapefiles contained in the Caravan dataset to derive the elevation zones.

For the catchments in the US and Great Britain, we used 1 October 1988 to 30 September 2013 as the simulation period, and
290 for the catchments in Brazil, we used 1 April 1988 to 31 March 2013 as the simulation period. The preceding five years were used as a warm-up period. Note that we did not distinguish between a calibration and validation period (i.e., we used the simulation period for calibration and evaluation) because we are interested in the influence of the different data types on model performance (cf. Tarek et al., 2020).

To account for equifinality, we calibrated the model for each scenario and catchment 100 times. From these 100 optimized
parameter sets and their corresponding simulated hydrographs, we calculated the ensemble mean hydrograph based on the arithmetic average of the 100 simulated streamflow values for each day. We compared this simulated hydrograph to the observed hydrograph to obtain one KGE value per data scenario for each catchment.

**Table 2: Overview of the seven combinations of calibration data used for the different scenarios of the modelling experiment. In**
**addition to the forcing data from the Caravan and the Camels datasets (see Sect. 3.2 and Table 1 for details), we also used the Hargreaves-based $E_{pot}$ values based on Caravan data as an alternative to the unrealistically high $E_{pot}$ data in the Caravan dataset (see Sect. 3.3.3).**

| Scenario | Scenario description | Precipitation | Temperature | $E_{pot}$ |
|---|---|---|---|---|
| I | Camels | Camels | Camels | Camels |
| II | Caravan | Caravan | Caravan | Caravan |
| III | Camels, but with Caravan precipitation data | Caravan | Camels | Camels |
| IV | Camels, but with Caravan temperature data | Camels | Caravan | Camels |
| V | Camels, but with Caravan $E_{pot}$ data | Camels | Camels | Caravan |
| VI | Camels, but with Hargreaves $E_{pot}$ data | Camels | Camels | Hargreaves |
| VII | Caravan, but with Hargreaves $E_{pot}$ data | Caravan | Caravan | Hargreaves |

## 4.2    Results

### 4.2.1    Model performances with Camels and Caravan data

Using the Camels forcing data for model calibration (scenario I) led to good model performances for most catchments (Fig. 6, Fig. 7). For the US catchments, the KGE ranged from 0.12 to 0.96 (median: 0.85) and for 20 of the 482 catchments (4 %), it was below 0.6. For the Brazilian catchments, the KGE ranged from -0.85 to 0.94 (median: 0.77); it was negative for two

catchments and below 0.6 for 52 of the 376 catchments (14 %). For the catchments in Great Britain, the KGE ranged from -

2.27 to 0.98 (median: 0.92); it was negative for three catchments, and below 0.6 for 13 of the 394 catchments (3 %). For the

five catchments with a negative KGE, the simulated streamflow was higher than the observed streamflow but the observed

streamflow was less than expected based on the precipitation and $E_{pot}$ data.

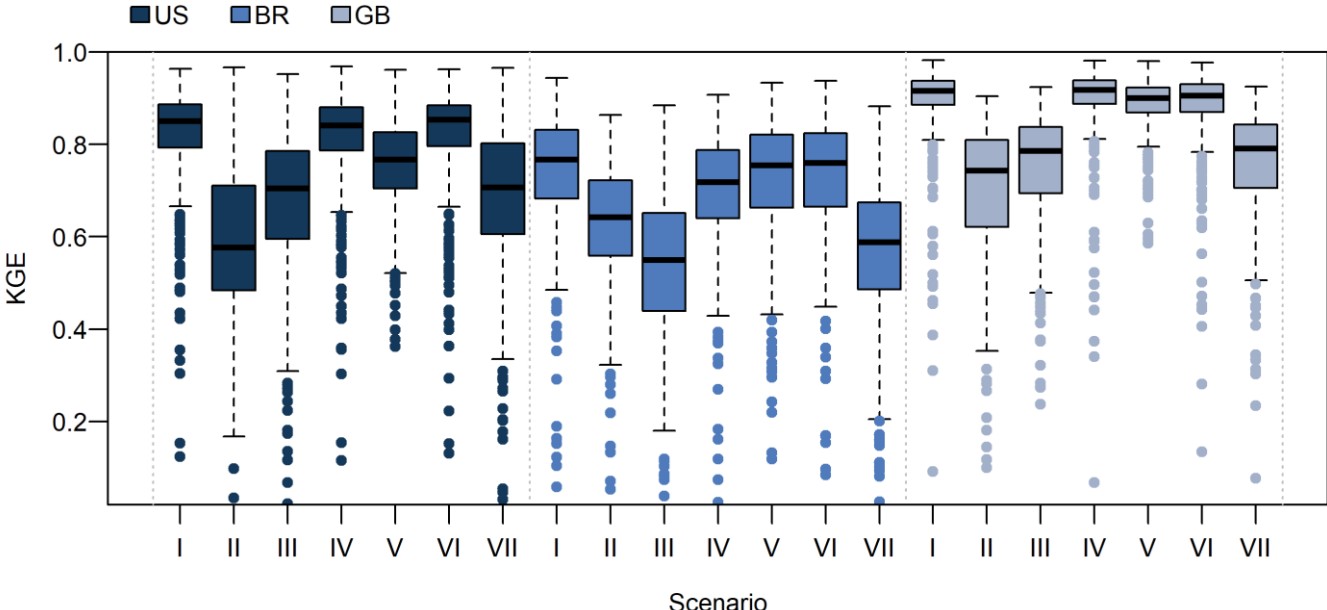

**Figure 6: Boxplots illustrating the model performances (KGE values for the ensemble mean hydrograph) for all scenarios (see Table 2 for a description) for all catchments in the US (n=482), Brazil (BR; n=376), and Great Britain (GB; n=394). The lower limit of each box represents the 25th percentile, the upper limit the 75th percentile, and the line the median. The whiskers end at the most extreme data point within 1.5 times the interquartile range. The dots represent outliers. Note that the y-axis was limited to positive KGE values. The KGE values were negative for 46 cases.**

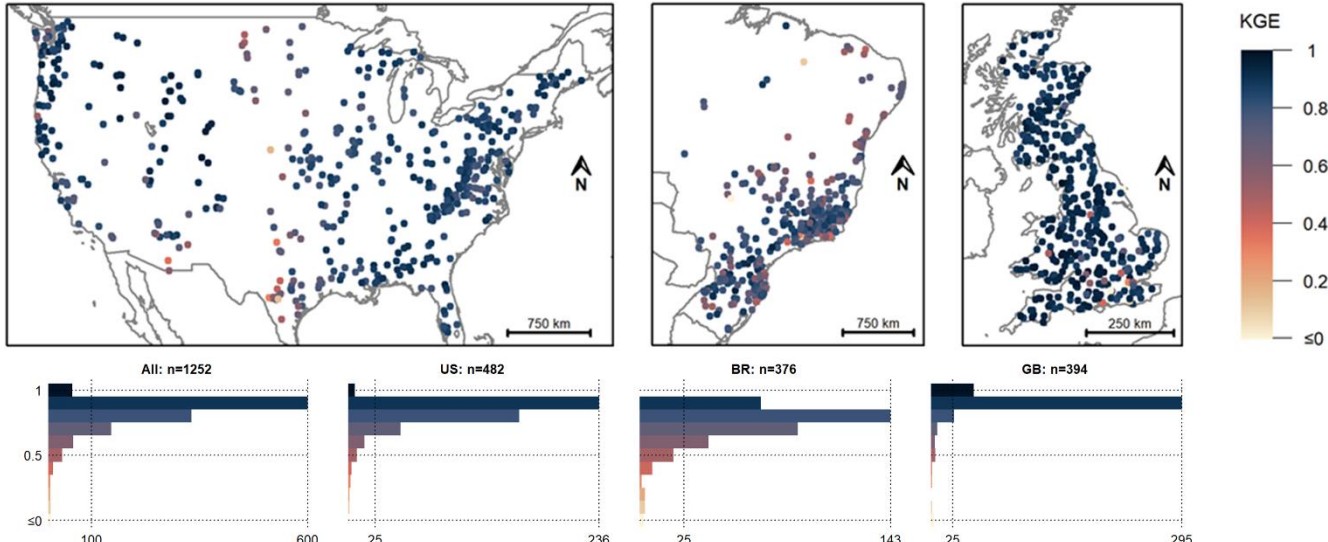

**Figure 7: Model performance (KGE values) for scenario I (Camels forcing data) for the catchments in the US, Brazil and Great Britain for the period April 1988 to March 2013 (Brazil) or October 1988 to September 2013 (US, Great Britain). Note that the lower limit of the scale was cut at 0. The KGE was negative for five catchments. The KGE values were rounded to one decimal for the histograms.**

Compared to calibration with the Camels data, calibration with the Caravan data (scenario II) decreased the KGE for 1134 of 1252 catchments (91 %; Fig. 6, Fig. 8, Table 3). The KGE for the calibration with Caravan data was below 0.6 for 488 of the 1252 catchments (39 %, i.e., 403 catchments more than for scenario I (Camels data)). However, the Caravan forcing data led to a positive KGE for all catchments, i.e., for the five catchments in Brazil and Great Britain for which the KGE for the calibration with the Camels data was negative, calibration with Caravan data resulted in positive KGE values. For these five catchments, the simulated streamflow was overestimated with the Camels forcing data and lower for the Caravan forcing data, and thus more similar to the observed streamflow.

**Table 3: Effect of differences in all forcing data (i.e., comparison of scenarios II and I), precipitation data (scenarios III and I), temperature data (scenarios IV and I), $E_{pot}$ data (scenarios V and I) from the Camels and Caravan datasets on model performance (i.e., KGE values), as well as effect of using Hargreaves $E_{pot}$ data instead of the $E_{pot}$ data from the Camels datasets (scenarios VI and I), or the Caravan dataset (scenarios VII and II) on model performance for all catchments together and each region. The stars indicate the statistical significance of the one-sided Wilcoxon test: \*\* indicates a p-value <0.001, \* indicates a p-value <0.01. For all tests except the effect of using the Hargreaves-based $E_{pot}$ data instead of the $E_{pot}$ data from the Caravan dataset, we tested for a significant decrease in model performance; for the latter, we tested for a significant increase in model performance (last column).**

| Effect of differences in: Comparison of scenarios: | | All data II-I | Precip. III-I | Temp. IV-I | $E_{pot}$ V-I | $E_{pot}$ VI-I | $E_{pot}$ VII-II |
|---|---|---|---|---|---|---|---|
| Median ΔKGE and significance | All (n=1252) | -0.17\*\* | -0.14\*\* | -0.00\*\* | -0.02\*\* | -0.00\*\* | 0.04\*\* |
| | US (n=482) | -0.25\*\* | -0.14\*\* | -0.00\*\* | -0.05\*\* | 0.00 | 0.10\*\* |
| | BR (n=376) | -0.11\*\* | -0.19\*\* | -0.03\*\* | -0.00\* | -0.01\*\* | -0.02 |
| | GB (n=394) | -0.17\*\* | -0.11\*\* | 0.00 | -0.02\*\* | -0.01\*\* | 0.08\*\* |
| Number and percentage of catchments with ΔKGE > 0.1 | All (n=1252) | 39 (3 %) | 23 (2 %) | 3 (0 %) | 61 (5 %) | 9 (1 %) | 423 (34 %) |
| | US (n=482) | 7 (1 %) | 3 (1 %) | 2 (0 %) | 9 (2 %) | 0 (0 %) | 245 (51 %) |
| | BR (n=376) | 17 (5 %) | 10 (3 %) | 1 (0 %) | 28 (7 %) | 2 (1 %) | 17 (5 %) |
| | GB (n=394) | 15 (4 %) | 10 (3 %) | 0 (0 %) | 24 (6 %) | 7 (2 %) | 161 (41 %) |
| Number and percentage of catchments with ΔKGE < 0 | All (n=1252) | 1134 (91 %) | 1169 (93 %) | 757 (60 %) | 855 (68 %) | 786 (63 %) | 433 (35 %) |
| | US (n=482) | 434 (90 %) | 443 (92 %) | 293 (61 %) | 385 (80 %) | 205 (43 %) | 84 (17 %) |
| | BR (n=376) | 333 (89 %) | 349 (93 %) | 339 (90 %) | 200 (53 %) | 275 (73 %) | 255 (68 %) |
| | GB (n=394) | 367 (93 %) | 377 (96 %) | 125 (32 %) | 270 (69 %) | 306 (78 %) | 94 (24 %) |
| Number and percentage of catchments with ΔKGE < -0.1 | All (n=1252) | 873 (70 %) | 770 (62 %) | 34 (3 %) | 236 (19 %) | 26 (2 %) | 172 (14 %) |
| | US (n=482) | 375 (78 %) | 287 (60 %) | 0 (0 %) | 160 (33 %) | 4 (1 %) | 21 (4 %) |
| | BR (n=376) | 208 (55 %) | 265 (70 %) | 34 (9 %) | 50 (13 %) | 16 (4 %) | 115 (31 %) |
| | GB (n=394) | 290 (74 %) | 218 (55 %) | 0 (0 %) | 26 (7 %) | 6 (2 %) | 36 (9 %) |

For the catchments in the US, the KGE mainly decreased for the catchments east of the 100° W meridian and along the West Coast. For the remainder of the western part of the US, the KGE did not change considerably. For the catchments in Brazil, the KGE tended to decrease most for the more southern catchments, but there were also some catchments in the eastern part of Brazil for which the KGE decreased quite strongly. The KGE increased for a few Brazilian catchments. For the catchments along the western coast of Great Britain, the KGE decreased strongly. The decrease was less strong for catchments in the southern part. For some catchments in southern England, the KGE increased (Fig. 8). This included a cluster of catchments for which the KGE was comparably low when calibrated with the Camels data (scenario I, Fig. 7).

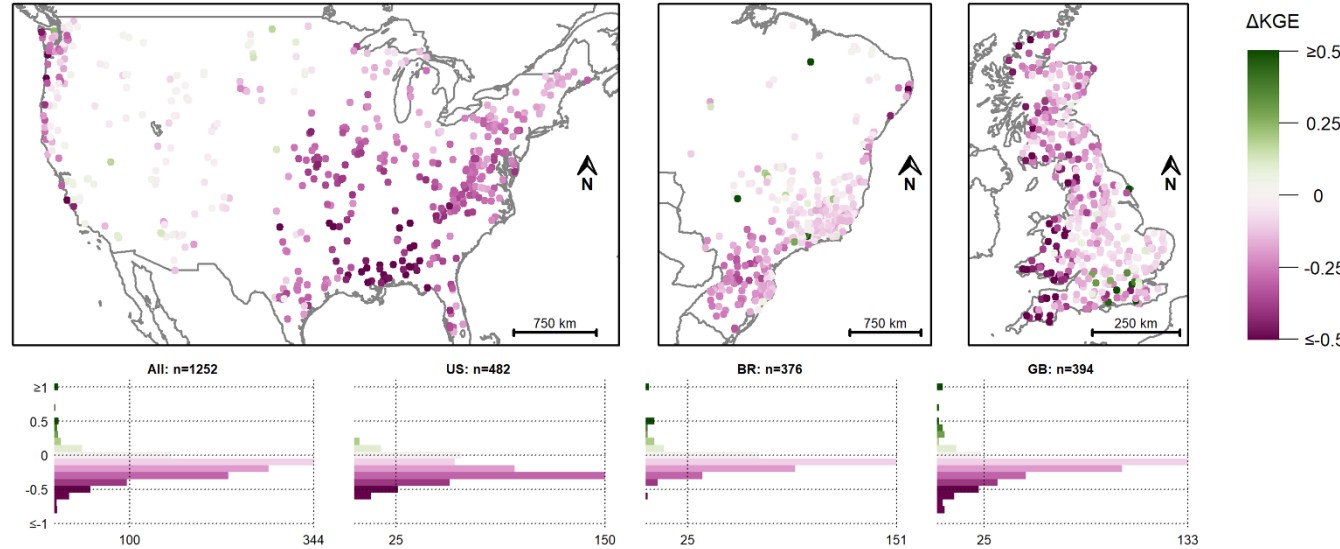

**Figure 8: Difference in model performance when using the Caravan forcing data (scenario II) and the Camels forcing data (scenario I). The pink colours indicate a lower KGE when calibrating with the Caravan data, the green colours indicate a higher KGE when calibrating with the Caravan data. Note that the colour scale was cut at a difference in KGE of ±0.5 and that the y-axes of the histograms were cut at a difference in KGE of ±1. The ΔKGE values were rounded to one digit for the histograms.**

### 4.2.2 Effect of differences in precipitation data

Model performances were higher when Camels precipitation data were used for model calibration than when Caravan precipitation data were used (Fig. 6). Using the Caravan precipitation data (scenario III) instead of the Camels precipitation data (scenario I) decreased the KGE for 1169 of the 1252 catchments (93 %) (Table 3). The pattern of the effect of the Caravan precipitation data on the KGE values was similar to the pattern of the effect of all the Caravan forcing data (Fig. 8). Indeed, the median difference between the KGE achieved with scenario II and scenario III for all 1252 catchments was -0.03, i.e., scenario III performed only slightly better than scenario II. The median difference was -0.09 for the US catchments, 0.06 for the Brazilian catchments (where scenario II performed better than scenario III, see Fig. 6 and Table 3), and -0.07 for the catchments in Great Britain. In other words, the difference in the precipitation data explained most of the effect of replacing the forcing data from the Camels datasets with the forcing data from the Caravan dataset. Furthermore, the effect of the difference in the precipitation data was larger than the effect of the difference in the temperature data and also the effect of the large difference in the $E_{pot}$ data (see Sect. 4.2.3 and 4.2.4).

### 4.2.3 Effect of differences in temperature data

The effect of using temperature data from the Caravan dataset (scenario IV) instead of temperature data from the Camels datasets (scenario I) was comparably small (Fig. 6). However, when considering all 1252 catchments, as well as when considering only the US catchments or only the Brazilian catchments, the KGE values still decreased significantly in scenario IV compared to scenario I (Table 3; $p<0.001$). Only in Great Britain, where the mean daily temperature data in the Caravan dataset were very similar to the mean daily temperature data in the CAMELS-GB dataset for most catchments (Fig. 2), no significant decrease in the KGE values was found when scenario IV was compared to scenario I ($p=1.0$), i.e., replacing the temperature data from the CAMELS-GB dataset with the temperature data from the Caravan dataset did not have a significant effect. There was no indication that the replacement of the temperature data had a stronger influence on the KGE for snow-dominated (mountainous) catchments than other catchments, as it may have been expected.

### 4.2.4 Effect of differences in potential evapotranspiration data

Using the $E_{pot}$ data from the Caravan dataset (scenario V) instead of the $E_{pot}$ data from the Camels datasets (scenario I), significantly decreased the KGE (Table 3; $p<0.01$ for the Brazilian catchments and $p<0.001$ for the catchments in the US and Great Britain, or when taking all catchments together). The decrease was particularly pronounced for the catchments in the US, where the differences between the mean annual $E_{pot}$ from the Caravan dataset and the mean annual $E_{pot}$ from the CAMELS-US dataset were especially large (Fig. 3, Fig. 5). However, compared to the KGE decrease when all forcing data were taken from the Caravan dataset (scenario II) or when only precipitation data were taken from the Caravan dataset (scenario III), the effect of the unrealistic $E_{pot}$ data from the Caravan dataset was relatively small (Fig. 6).

The model performance drop compared to scenario I tended to be smaller when the model was calibrated with the Hargreaves $E_{pot}$ data (scenario VI), than when the model was calibrated with the $E_{pot}$ data from the Caravan dataset (scenario V). For the US catchments, there was no significant decrease in KGE when the $E_{pot}$ data calculated with the Priestley-Taylor equation for the CAMELS-US dataset were replaced with the Hargreaves $E_{pot}$ data (compare scenario VI to scenario I; $p=0.987$; Table 3), while this was the case when replacing the $E_{pot}$ data from the Camels datasets with the Hargreaves $E_{pot}$ data for Brazil and Great Britain ($p<0.001$).

Similarly, we tested whether replacing the ERA5-Land $E_{pot}$ data in the Caravan dataset with the Hargreaves $E_{pot}$ data (scenario VII) significantly improved the model performance compared to when all forcing data from the Caravan dataset were used (scenario II). This was indeed the case ($p<0.001$) when either all catchments, all US catchments, or all catchments in Great Britain were considered (Table 3, last column). However, for the Brazilian catchments, the effect was the opposite, i.e., the unrealistic $E_{pot}$ data from the Caravan dataset led to significantly better results than the alternative Hargreaves $E_{pot}$ data ($p<0.001$).

A positive effect of the Hargreaves $E_{pot}$ data instead of the $E_{pot}$ data from the Caravan dataset on the model performance could be observed especially for regions in which the use of Caravan forcing data (scenario II) instead of Camels forcing data (sce-

400 nario I) had a strong negative impact (Fig. 8, Fig. 9). In the US, this was mainly the case for the catchments in the eastern part of the country and along the West Coast. The few catchments in Brazil for which the model performance increased due to the Hargreaves $E_{pot}$ data were located in the southern part of the country, as well as along the eastern coast. In Great Britain, the increases tended to be stronger in the western part of the country (Fig. 9).

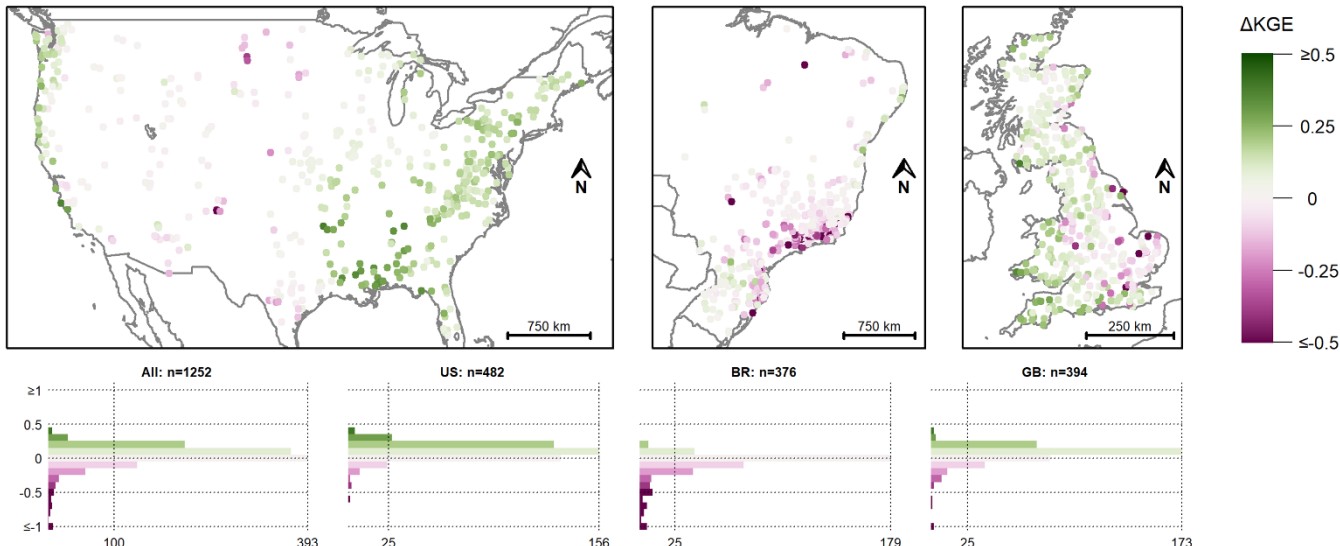

**Figure 9: Difference in KGE values for the model calibration with precipitation and temperature data from the Caravan dataset and the Hargreaves-based $E_{pot}$ data (scenario VII) and the calibration when all Caravan forcing data were used (scenario II). The pink colours indicate a lower KGE value when the Hargreaves-based $E_{pot}$ data were used compared to the calibration with all Caravan data, the green colours indicate a higher KGE value with the Hargreaves-based $E_{pot}$ data. Note that the colour scale was**

410 **cut at a difference in KGE of ±0.5 and that the y-axes of the histograms were cut at a difference in KGE of ±1. The ΔKGE values were rounded to one decimal for the histograms.**

### 4.3 Discussion of the difference in model performances for Camels and Caravan data

Streamflow modelling with the forcing data included in the three Camels datasets worked well for most catchments. An un-

415 suitable model structure, errors in the Camels data, or human impacts on streamflow are possible explanations for the poor model performances for some of the catchments. For example, the catchments in the arid regions of the US for which the model performances were low were identified as more difficult to model in earlier studies as well (Knoben et al., 2020; Kollat et al., 2012). Based on the comparison of different models, Knoben et al. (2020) found that there are model structures that can simulate the discharge in these catchments successfully. Similarly, the low model performances for some catchments in south-

420 eastern Great Britain may be attributed to complex groundwater systems (as identified earlier by Lane et al., 2019; Seibert et

al., 2018). A more suitable model structure accounting for subsurface losses would lead to a better model performance (Kiraz et al., 2023). However, looking at the model results as an aggregated measure of data quality, the good model fits indicate a high data quality of the data in the CAMELS-US, CAMELS-BR, and CAMELS-GB datasets.

The overall deterioration in model performances for calibration with the Caravan dataset indicates that the quality of the forcing data from ERA5-Land is lower than the quality of the data that are available for the US, Brazil, and Great Britain. As the ERA5-Land data are coarser than most data in the Camels datasets (Table 1), this was to a certain extent expected. Furthermore, the negative effect of the Caravan forcing data on model performance may be smaller for models that are less sensitive to errors in the input data and can adapt more flexibly. However, a user who decides to use the Caravan dataset instead of different Camels datasets out of convenience may not be aware of the considerable degradation of the input data and the potentially severe effects on the model performance.

Even though the $E_{pot}$ data in the Caravan dataset are unrealistically high for many catchments (Fig. 3), the analysis of the isolated effects of the Caravan forcing data showed that differences in the precipitation (Fig. 1) were responsible for most of the decrease in model performance for the Caravan forcing data (Fig. 6, Table 3). As precipitation is the main driver of stream-flow, the strong influence of precipitation is not surprising. That the model performance dropped so much indicates more than a small bias and rather a lower plausibility of the reanalysis-based precipitation data (cf. Beck et al., 2017; Tarek et al., 2020; Wang et al., 2023a).

The spatial differences in how the model performance was affected by the Caravan precipitation data may be related to both the spatial patterns in the errors and the catchment characteristics. For example, the Caravan precipitation data led to a much stronger deterioration in model performance for catchments in the eastern part of the US than in the western part. This pattern was also observed in earlier studies that tested the value of reanalysis data for hydrological modelling in North America (Essou et al., 2016; Tarek et al., 2020). Essou et al. (2016) mainly attributed this issue to the convective summer storms in the eastern part of the US that are poorly represented in the reanalysis data.

These results mean that one should be cautious when using the Caravan precipitation dataset instead of more reliable (e.g., station-based) precipitation data because the conclusions may be affected by the lower data quality of the forcing data in the Caravan dataset. In our opinion, ERA5-Land precipitation data should only be used for catchments for which there are no alternative data (so that these catchments can still be included in large-sample studies). This is in line with the conclusions of Essou et al. (2016, 2017) and Tarek et al. (2020), who stated that reanalysis data can serve as a proxy for meteorological data for regions with little or no weather station data.

Considering the large bias of the Caravan $E_{pot}$ data (Fig. 3, Fig. 5), the effect on the model performance was surprisingly small and clearly smaller than the effect of the precipitation data (Fig. 6, Table 3: III-I versus V-I). This is in line with earlier studies that showed that $E_{pot}$ data affect model performance less than precipitation data (Oudin et al., 2006; Paturel et al., 1995) because the model can compensate for a systematic overestimation of $E_{pot}$. Thus, an overestimation of $E_{pot}$ is less severe than an under-estimation (cf. Jayathilake and Smith, 2022). Indeed, additional sensitivity analyses with artificially biased $E_{pot}$ data, not shown here for the sake of brevity, showed that the HBV model compensated for the overestimated $E_{pot}$ data from the Caravan dataset

mainly by adjusting the values of the parameters of the soil routine to reduce evapotranspiration. This allowed the model to simulate an actual evapotranspiration that was more realistic and of a similar order of magnitude as the actual evapotranspiration simulated with the $E_{pot}$ data from the Camels datasets. Thus, even though the model performance may have not changed considerably, the processes were represented differently due to the compensation. This is problematic, especially when the calibrated parameter values are subsequently used to characterize a catchment (cf. Bouaziz et al., 2022).

The few cases for which the model performance was better with the $E_{pot}$ data from the Caravan dataset can either be attributed to even more (but exceptional) erroneous Camels data or to compensation effects of biased variables (cf. Wang et al., 2023b). Possible explanations for the catchments for which the unrealistically high $E_{pot}$ data led to an increase in model performance may be a wrong representation of the processes that coincidentally led to a better model performance (Kirchner, 2006) or errors in the water balance data for the Camels dataset and thus an improvement thanks to the high (but still wrong) $E_{pot}$ data. A compensation of the wrong water balance with overestimated $E_{pot}$ data may also explain why many catchments in Brazil did not profit from the more realistic Hargreaves-based $E_{pot}$ data (cf. Fig. 9).

The low sensitivity of the hydrological model to the wrong $E_{pot}$ data indicates that validating meteorological forcing data with a hydrological model approach, as we did in this study, may not be the most suitable way to investigate the quality of $E_{pot}$ data but works fine for precipitation data. Thus, other approaches or simple plausibility tests may be more useful for the validation of $E_{pot}$ data and the indices calculated thereof.

## 5    Suggestions for use of the Caravan dataset

For the vast majority of the catchments, using the forcing data from the Caravan dataset deteriorates model results and impacts the conclusions drawn from them. In our opinion, the model performances were affected so strongly by the use of the reanalysis data in the Caravan dataset that it cannot be considered an inconsequential trade-off between the use of homogeneous data and a drop in model performance. Even though we agree that the use of ERA5-Land data for all catchments has advantages, such as comparability and the possibility to extend the Caravan dataset to other catchments, the loss in data quality for this standardization is a hefty price tag.

Because the Caravan dataset is easy to acquire, well-organized, and offers opportunities for catchments in underrepresented regions to be included in large-sample studies in hydrology, there are clear advantages of using reanalysis data for some studies, and in particular for catchments for which the forcing data would otherwise not be available. The use of the Caravan dataset as the standard resource for large-sample hydrology would also facilitate the comparison of model results. However, the quality of the meteorological data that are used for hydrological model calibration is lower for the Caravan dataset than for the original Camels datasets. Thus, in our opinion, the Caravan forcing time series are not the most suitable dataset for all studies, in particular for catchments for which higher quality data are available. Therefore, we provide two suggestions to improve the Caravan dataset.

## 5.1 Extension with forcing data from the original datasets

To make researchers aware that they are using lower quality data when downloading the data from the Caravan dataset (than when they would use the Camels datasets), we suggest extending the Caravan dataset by also adding the forcing data that were originally included in the national and regional large-sample datasets when these are available. In this way, users would be able to decide if either global comparability or the use of the best possible data is more important for their study. Including both data types in Caravan would also lead to more transparency regarding the differences between the forcing data in the Camels datasets and the reanalysis data in the Caravan dataset. For catchments for which no other data are available than those from ERA5-Land, i.e., for which the ERA5-Land data are state of the art, no extension would be necessary. Of course, users already have this choice, since the Camels datasets are always available in their own repositories. Still, it would be much more convenient for the users finding them in Caravan, for facilitating their use and the comparison with the ERA5-Land data.

Until the Caravan dataset is extended in such a way, we highly recommend that users assess thoroughly if they want to use the Caravan dataset or if they prefer the data that were originally included in the Camels datasets. Especially if a study is limited to catchments for which better data are available, it may be valuable to go the more tedious way and download the different Camels datasets separately. Even though the Camels data are also not perfect, their quality is better than the one of the standardized data currently available in the Caravan dataset.

There are, of course, also situations in which the global comparability (and thus the reliance on input data that was generated uniformly for all catchments) is most important. In such cases using the Caravan forcing data is the best possible solution (at least currently), and we suggest using the Caravan data as it is (for all variables, except $E_{pot}$, see Sect. 5.2), even though this may mean a loss in data quality and model performance (which is larger for some catchments than for others). However, in most applications, we think that it is better to use the best possible data, as one would do in every other situation in life.

As an alternative or addition to the extension of the Caravan dataset with the original Camels data, a clear warning on the loss of data quality due to the standardization of the meteorological forcing data in the Caravan dataset is needed to avoid that the Caravan dataset is used instead of the original Camels datasets without the user being aware of the consequences. Such a warning would avoid duplicating already existing data and still enable the user to make an informed decision.

It can be considered a general lesson learned from this study that new large-sample datasets need to clearly state their advantages compared to already existing datasets, but also inform users about possible drawbacks. With ERA6, the next generation of reanalysis data is currently being developed. Considering the development of reanalysis data so far, it is expected that the quality will increase. This could change the appropriateness of reanalysis data as forcing data in hydrological models. However, if limitations of a new dataset are already known on beforehand, a disclaimer section in the accompanying publication should be added and the users should be informed about the limitations in the database itself. Furthermore, if issues with some of the data only become clear at a later point in time, this information should be added to the database. With that, it can be promoted that the right datasets are used for the right purposes.

## 5.2 Replacement of the ERA5-Land derived potential evapotranspiration data

The comparison of the $E_{pot}$ data included in the Caravan dataset with the $E_{pot}$ data from the CAMELS-US, CAMELS-BR, and CAMELS-GB datasets showed that the Caravan $E_{pot}$ data are systematically too high and are not reliable for any hydrological application. Because hydrological models can cope with some errors in the $E_{pot}$ input data (Andréassian et al., 2004; Bai et al., 2016; Oudin et al., 2006), we expect that this large difference is mainly problematic for the attributes based on these $E_{pot}$ data, such as the aridity index (see Fig. 4). Therefore, we suggest replacing the $E_{pot}$ data from ERA5-Land with an alternative method and recalculating the values of the catchment attributes that include the $E_{pot}$ data. The Hargreaves-based approach (see Sect. 3.3.3) is a possible alternative for the $E_{pot}$ data that could be included in Caravan. The advantages are that they are realistic and can be calculated based on the other ERA5-Land derived data (temperature and precipitation) that are already in the Caravan dataset. However, there are other methods to estimate $E_{pot}$ as well and different global datasets containing $E_{pot}$ estimates, such as the dataset presented by Singer et al. (2021) resulting from the application of the FAO's Penman-Monteith equation based on ERA5-Land meteorological variables. With Caravan being a community effort, making a suitable choice for new Caravan $E_{pot}$ data can be considered a task of the large-sample hydrology community. Aside from replacing the current $E_{pot}$ data with other globally available $E_{pot}$ data, our suggestion of including the forcing data from the original Camels datasets where possible as an alternative to the standardized global data (see Sect. 5.1) also applies for $E_{pot}$ data.

## 6 Conclusions

Currently, the Caravan dataset is the most comprehensive large-sample dataset available in hydrology. It provides the community with hydrometeorological information and catchment attributes for many catchments in the world and offers the opportunity to extend the dataset with catchments for which streamflow data (but potentially no meteorological data) are available. It, furthermore, allows the forcing data to be comparably easily updated. Therefore, the Caravan dataset brings large-sample hydrology to the next level. However, there are considerable differences between the forcing data included in the Caravan dataset and the forcing data in the original large-sample datasets, as shown here for the CAMELS-US, CAMELS-BR, and CAMELS-GB datasets. The goal of this paper is to make researchers aware of these differences and to show that these differences cause a reduction in model performance for most catchments. The impact of the lower quality data on model results may lead to wrong conclusions, for example regarding the suitability of a model or its parametrization. It can also affect conclusions regarding the suitability of regionalization approaches and the value of data for calibration of otherwise ungauged catchments. Therefore, we suggest that the standardized global forcing data in the Caravan dataset are extended with the higher quality forcing data from the original data sources where available. We also suggest using other $E_{pot}$ data, e.g., calculated from the temperature data included in the Caravan dataset, as the ERA5-Land $E_{pot}$ data are unrealistically high for many catchments. Even though this does not affect the model calibration results as much as the differences in the precipitation data, it can lead to wrong parameterizations and affects the catchment attributes (and thus catchment comparisons). We are sure that these

relatively easy changes will increase the value of the Caravan dataset further and support its establishment as the main resource for large-sample hydrology.

## 7    Code and data availability

The Caravan dataset (Kratzert et al., 2023b) is available from Zenodo, with doi: 10.5281/zenodo.7944025. The CAMELS-US dataset (Addor et al., 2017a; Newman et al., 2014) is available from https://ral.ucar.edu/solutions/products/camels (last accessed: March 6, 2024). The CAMELS-BR dataset (Chagas et al., 2020b) is available from Zenodo, doi: 10.5281/zenodo.3964745. The CAMELS-GB dataset (Coxon et al., 2020b) is available from the NERC Environmental Information Data Centre, doi: 10.5285/8344e4f3-d2ea-44f5-8afa-86d2987543a9.

The HBV model in the version HBV-light is available from https://www.geo.uzh.ch/en/units/h2k/Services/HBV-Model/HBV-Download.html (last accessed: March 6, 2024).

An R script for the calculation of the Hargreaves-based $E_{pot}$ values is available from Zenodo, doi: 10.5281/zenodo.10784701. All colourmaps used in this paper are scientific colourmaps from Crameri (2023), accessed via the R package "scico", version 1.5.0 (Pedersen and Crameri, 2023). Other R packages used for this study are "circlize", version 0.4.15 (Gu et al., 2014); "hydroGOF", version 0.4-0 (Zambrano-Bigiarini, 2023); "vioplot", version 0.4.0 (Adler et al., 2022); "rworldmap", version 1.3-6 (South, 2011); "rworldxtra", version 1.01 (South, 2012); and "maps", version 3.4.1 (Becker et al., 2022).

## 8    Author contribution

FCS: Conceptualization; Data curation; Formal analysis; Investigation; Methodology; Validation; Visualization; Writing – original draft preparation; Writing – review and editing

GS: Conceptualization; Data curation; Formal analysis; Investigation; Validation; Writing – review and editing

MN: Supervision; Writing – review and editing

ET: Supervision; Writing – review and editing

IvM: Conceptualization; Methodology; Resources; Supervision; Writing – review and editing

JS: Conceptualization; Methodology; Resources; Supervision; Writing – review and editing

## 9    Competing interests

ET and JS are members of the editorial board of Hydrology and Earth System Sciences.

## 10 Acknowledgements

We thank Marc Vis for providing the scripts to create the catchment files for the HBV model, for his advice on parallel computing, and previous discussions on hydrological modelling. We thank the Science IT team of the University of Zurich (www.s3it.uzh.ch) for providing the infrastructure for cloud computing. We thank Nans Addor and Frederik Kratzert for their helpful comments on an earlier version of this manuscript. We also thank the editor and the two reviewers for their constructive comments.

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
