# Peer review of "HESS Opinions: A few camels or a whole caravan?"

_EGUsphere, 2024_

## Author Comment (AC2)

Dear Reviewer 2

Many thanks for your valuable comments on our manuscript. We were happy to read that you appreciated the paper. Please find below our replies to the comments and how we will implement them in the revised version of the paper. We used *blue italic font* for the comments and black font for our replies.

Best regards,

Franziska Clerc-Schwarzenbach on behalf of all co-authors

**General comments**

*RC: "As a strong believer in the importance of large sample studies in hydrology, I read the paper with much interest. I found the results very interesting despite being unsurprised by the results. The strengths and drawbacks of ERA5 and its little brother ERA5-Land data are fairly well known, especially when it comes to temperature (very good) and precipitation (good with some issues such as regional biases). Potential evapotranspiration is more of an unknown, and I found its use in Caravan a bit perplexing since it is an unknown quantity. I believe that 'exotic' data from reanalysis should be thoroughly validated before their incorporation into any hydrological study."*

We are happy to read that you found the paper interesting. We agree that the results can be considered unsurprising as one cannot expect global reanalysis data to be as good as station-based (or regional) data. However, we think that the impact of these differences on the model results can still be surprising to many users of the Caravan dataset and that the poor potential evapotranspiration data may surprise people who assume that the data in a large-sample dataset has a fair quality, as you also mention.

*RC: "My understanding is that potential evapotranspiration is computed using the surface energy balance assuming a crop soil surface, as it was included for irrigation purposes. As such, it is not surprising that catchment scale estimates would be severely overestimated I many cases. See Muñoz-Sabater et al. (2021) for example."*

The definition of potential evapotranspiration in ERA5-Land changed in November 2021 (as stated on the website, see the first link below, under "Known issues / Definition of Potential Evaporation (PEV) modified"). While Muñoz-Sabater (2021) indeed write that for the vegetation type crops were assumed and that it was also assumed that there was no soil moisture stress, the ERA5-Land documentation states that the computation of potential evapotranspiration in ERA5-Land assumes an open water surface (i.e., pan evaporation) and that the atmosphere is not affected by it. This difference between ERA5 and ERA5-Land is also stated elsewhere in the documentation (see the first link below, under "Guidelines / Actual and potential evapotranspiration"). It is furthermore confirmed in the variable description where the data can be downloaded (see the second link).

- First link: https://confluence.ecmwf.int/display/CKB/ERA5-Land%3A+data+documentation
- Second link: https://cds.climate.copernicus.eu/cdsapp#!/dataset/reanalysis-era5-land

We will clarify this difference between ERA5 and ERA5-Land in our revised manuscript.

*RC: "Despite calling the results 'unsurprising', I believe this paper makes some very good observations that are useful to the community [...]."*

Thank you very much for listing our main findings and summarizing why our manuscript is helpful for the community. It was valuable to see that the points that we wanted to make came across clearly.

**Specific comments**

*RC: "I am not fully sure why this is considered an Opinion paper. To me, the breadth of the research work and analysis clearly qualifies it as a research paper. I did not see much 'opinion' in this paper, as most of the arguments/discussions are results-based. Consequently, I suggest this submission be reclassified as a research paper."*

Thank you for sharing your thoughts on the category of this paper. Originally we were planning to write a shorter paper and therefore contacted the editors about writing an opinion paper. However, we agree that the paper has evolved in a more detailed analysis that is beyond a typical opinion paper. We would be happy to change the category, but leave the decision to the editor.

*RC: "I think the title does a disservice to the paper. I like the catchy phrase, but in reality, I would think that a majority of hydrologists are not familiar with Camel, and an even smaller number are aware of Caravan. A more generic title referring to large sample datasets and global datasets would be more appropriate for the varied readership of HESS."*

The main target group for this paper are potential users of the Caravan dataset that are aware of this dataset and the Camels datasets. Thus, we like to keep these two word in the title to grab the attention of the target group. However, we will change the title to "Large-sample hydrology: A few camels or a whole caravan?" to provide potential readers with some more information on the topic of the paper.

*RC: "I believe an additional discussion point should be added regarding the choice of a particular hydrological model. It is well known that some models may be more flexible than others at adapting to biases in input variables such as precipitation and easily scale PET with specific calibration parameters. This is mentioned in the paper, but I believe some other hydrological models may perform better than the one used in this study, and the performance drop mentioned in this study may not be as bad. I certainly would not expect the conclusions of this paper to be any different, but this should be mentioned."*

We will add to section 4.3 that it is possible that the effect of the Caravan forcing data is different (either smaller or larger) when a different model is used as the sensitivity to the input data may be model dependent. We also do not think that the results will be very different for a different lumped conceptual model model but agree that it would be interesting to look deeper into the various effects of the choice of the input data for different models.

*RC: "There should be a mention of the upcoming ERA6 reanalysis. The ERA5 reanalysis used in Caravan will soon be a thing of the past. In addition to improved resolution, ERA6 will have a full overhaul of the model physics, including radiation, which is overestimated in ERA5 and likely part of the PET problem, in addition to the issue discussed above. Based on past history, we can expect a significant performance increase with ERA6. This should be mentioned in the paper. I believe that reanalysis is indeed the future of large-sample hydrology and that merging reanalysis with Deep Learning approaches will produce very high-quality global datasets much sooner than most people think. Already, the merging of deep-learning methods with weather forecasting models promises to revolutionize weather forecasting—exciting times."*

Thank you for pointing this out. We will add to section 2 that the ERA6 dataset is currently in development and that a quality increase can be expected. While we could not find any papers about ERA6 that are available already, we agree that it is important to show that comparisons of reanalysis data and other data may lead to other results in the future.

*RC: "I would also like to add one important advantage of global datasets based on reanalysis that was not mentioned in the paper: they are easily updated once a new version comes out. In addition, new data is produced in near-real time. Comparatively, datasets relying on observations (e.g., Camel) are much more complex to update (missing data, stations being decommissioned, etc.) and, based on past history, are unlikely to be updated at all, or very infrequently. A dataset such as Caravan will still need to be updated, but the process is much more straightforward."*

We will add to section 1 that the possibility to simply update a large-sample dataset based on reanalysis data is an advantage of datasets that use globally available forcing data. We will take up this point again in the conclusions (section 6) as well. However, we also want to stress the point that having datasets that remain the same over longer periods has advantages; mainly, that it allows for a better comparison between different studies using the same data.

*RC: "The use of 'significant/ly' should be clarified if it is in the 'statistical' sense from the get-go at line 70. In some cases, it clearly is, but not so much in others."*

Thank you for making us aware that it is not always clear whether we refer to a statistical significance when we use this expression or to a considerable difference. Indeed, when using the expression for the first time in line 70, it was unconnected to a statistical test. Therefore, we will change the wording to "considerably". We assume that in Table 3 and the corresponding caption, it is clear that we mean a statistical significance. However, we will add the word "statistical" before "significance". For the two occurrences in paragraph 4.2.3 and the occurrence in paragraph 4.2.4 for which we did not mention the corresponding p-value, we will add the p-values to clarify the statistical significance.

*RC: "I would suggest the use of PET instead of $E_{pot}$, with the former being a lot more common, in my opinion."*

Even though we see the wide use of PET, we prefer to use (and keep) $E_{pot}$ in this manuscript. As we are already using a lot of abbreviations (all consisting of several capital letters) for the datasets, we think that it is better to use the variant with a subscript for the potential evapotranspiration to make it more distinct. Furthermore, $E_{pot}$ can be used in the formula for the aridity index, whereas the use of PET in an equation would be mathematically incorrect (as it would equal P times E times T).

---

## Author Response (AR1)

Dear Editor, dear Thom

Thank you very much for your encouraging comments on our manuscript. We also appreciated the valuable feedback by the reviewers. Please find below the comments that suggested changes to the manuscript and how we addressed them. We used *blue italic font* for the comments by the reviewer 1, *purple italic font* for the comments by the reviewer 2, black font for our replies. We sorted the comments by the order of occurrence of the changes in the revised manuscript. In addition, we made some very minor changes to the manuscript to improve clarity or readability and avoided colouring of table cells.

Best regards,

Franziska Clerc-Schwarzenbach on behalf of all co-authors

*RC: "I am not fully sure why this is considered an Opinion paper. To me, the breadth of the research work and analysis clearly qualifies it as a research paper. I did not see much 'opinion' in this paper, as most of the arguments/discussions are results-based. Consequently, I suggest this submission be reclassified as a research paper."*

Thank you for sharing your thoughts on the category of this paper. Originally, we were planning to write a shorter paper and therefore contacted the executive editors about writing an opinion paper. However, we agree that the paper has evolved in a more detailed analysis that is beyond a typical opinion paper. We would be happy to change the category, but are also fine with staying with an opinion paper if the editor would prefer that.

*RC: "I think the title does a disservice to the paper. I like the catchy phrase, but in reality, I would think that a majority of hydrologists are not familiar with Camel, and an even smaller number are aware of Caravan. A more generic title referring to large sample datasets and global datasets would be more appropriate for the varied readership of HESS."*

The main target group for this paper are potential users of the Caravan dataset. They are thus aware of this dataset and the Camels datasets. Thus, we like to keep these two words in the title to grab the attention of the target group. However, we changed the title to "Large-sample hydrology – A few camels or a whole caravan?" to provide potential readers with some more information on the topic of the paper. If the editor decides to continue to classify the paper as an opinion paper, we suggest "HESS Opinions: Large-sample hydrology – A few camels or a whole caravan?" as title.

*RC: "I would also like to add one important advantage of global datasets based on reanalysis that was not mentioned in the paper: they are easily updated once a new version comes out. In addition, new data is produced in near-real time. Comparatively, datasets relying on observations (e.g., Camel) are much more complex to update (missing data, stations being decommissioned, etc.) and, based on past history, are unlikely to be updated at all, or very infrequently. A dataset such as Caravan will still need to be updated, but the process is much more straightforward."*

We added text to highlight that the possibility to simply update a large-sample dataset based on reanalysis data is an advantage of datasets that use globally available forcing data. However, we also want to stress the point that having datasets that remain the same over longer periods has advantages; mainly, that it allows for a better comparison between different studies using the same data.

We added this point in lines 56-57 in the revised manuscript.

RC: *"The use of 'significant/ly' should be clarified if it is in the 'statistical' sense from the get-go at line 70. In some cases, it clearly is, but not so much in others."*

Thank you for making us aware that it is not always clear whether we refer to a statistical significance when we use this expression or to a considerable difference. Indeed, when using the expression for the first time in line 70, it was unconnected to a statistical test. Therefore, we changed the wording to "considerably". We assume that in Table 3 and the corresponding caption, it is clear that we mean a statistical significance but added the word "statistical" before "significance" to make this even clearer. For the two occurrences in paragraph 4.2.3 and in paragraph 4.2.4 for which we did not mention the corresponding p-value, we added the p-values to clarify the statistical significance.

We made the changes in lines 71, 338, 372, and 380-381 in the revised manuscript.

RC: *"My understanding is that potential evapotranspiration is computed using the surface energy balance assuming a crop soil surface, as it was included for irrigation purposes. As such, it is not surprising that catchment scale estimates would be severely overestimated I many cases. See Muñoz-Sabater et al. (2021) for example."*

The definition of potential evapotranspiration in ERA5-Land changed in November 2021 (as stated on the website, see the first link below, under "Known issues / Definition of Potential Evaporation (PEV) modified"). While Muñoz-Sabater (2021) indeed write that for the vegetation type, crops were assumed and that it was also assumed that there was no soil moisture stress, the ERA5-Land documentation states that the computation of potential evapotranspiration in ERA5-Land assumes an open water surface (i.e., pan evaporation) and that the atmosphere is not affected by it. This difference between ERA5 and ERA5-Land is also stated elsewhere in the documentation (see the first link below, under "Guidelines / Actual and potential evapotranspiration"). It is also confirmed in the variable description where the data can be downloaded (see the second link).

- First link: https://confluence.ecmwf.int/display/CKB/ERA5-Land%3A+data+documentation
- Second link: https://cds.climate.copernicus.eu/cdsapp#!/dataset/reanalysis-era5-land

We added a clarification for the different ways of calculating potential evapotranspiration in ERA5 and ERA5-Land in lines 71-75 in the revised manuscript. We added the link to the ERA5-Land documentation, chapter "Known issues" to the reference list.

RC: *"Are the are evaluations of ERA5-Land reanalysis dataset outside the use for hydrological modelling that might have relevant insights into regional differences? The studies currently cited seem largely focused on hydrological application though I assume there must also be other uses of this dataset?"*

We looked into studies evaluating the ERA5-Land dataset. There are indeed quite some studies that evaluated temperature, precipitation, or both variables by comparing them to station data or satellite data for a certain region (Table 1). One study also evaluated potential evapotranspiration calculated based on different ERA5-Land variables (Vanella et al., 2022).

We added a paragraph (lines 89-103 in the revised manuscript) to summarize the findings presented in Table 1.

*Table 1: Summary of the findings of studies evaluating ERA5-Land data.*

| Reference and Location | Findings regarding ERA5-Land data |
| --- | --- |
| Almeida and Coelho (2023) Portugal | *Temperature*: Better performance in flat regions than in more complex terrain |
| Araújo et al. (2022) Northeast Brazil | *Temperature*: Good correlation with data from weather stations |
| Zhao and He (2022) Chinese Qilian mountains | *Temperature*: Good correlation with data from weather stations; trends represented well |
| Yilmaz (2023) Turkey | *Temperature*: Lower than from weather stations; trends represented well |
| Vanella et al. (2022) Italy | *Temperature*: Good correlation with data from weather stations
*Potential evapotranspiration*: Calculation of potential evapotranspiration based on other ERA5-Land variables reasonable (not potential evapotranspiration from ERA5-Land) |
| Gomis-Cebolla et al. (2023) Spain | *Precipitation*: Overestimation of light and moderate precipitation, underestimation of heavier precipitation; spatial and temporal patterns represented well; complex precipitation patterns and precipitation over complex terrain most problematic |
| Wu et al. (2023) Tibetan Plateau | *Precipitation*: Overestimation of light rain causing overall wet bias; spatial and temporal patterns represented well |
| Xie et al. (2022) China | *Precipitation*: Better representation of large-scale precipitation systems in NE than patterns in SW; tendency for wet bias |
| Tan et al. (2023) Kelantan basin (Malaysia) | *Temperature*: Underestimation of daily maximum temperatures, overestimation of daily minimum temperatures
*Precipitation*: Overall dry bias due to underestimation of strong precipitation events (but overestimation of moderate events); good correlation of monthly data with weather stations, only moderate correlation of daily data with weather stations
*Climate patterns*: More successful representation along the coast than in the mountains further inland |

*RC: "(Section 4.2) HBV and HyMod have been calibrated to the MOPEX catchments (precursor of CAMELS-US) with NSE (no KGE then) to identify problematic catchments (Kollat et al., 2012, WRR, doi:10.1029/2011WR011534). This might be a possible comparison of difficult to model catchments."*

and *RC: "(Section 4.3) The low performance of models like HBV in chalk catchments in the south of the UK is significantly reduced when a more suitable model structure for groundwater processes used. See the recent study by Kiraz et al. (2023, HSJ, https://doi.org/10.1080/02626667.2023.2251968) – results for KGE are in the supplemental material of the study."*

Thank you for your suggestion to compare our rather "problematic" catchments in the United States and Great Britain and the additional references. We added a reference to these studies, as well as to the studies by Lane et al. (2019) and Knoben et al. (2020) to the first paragraph of section 4.3 We implemented the changes on lines 416-421 in the revised manuscript. The four references (Kiraz et al., 2023; Knoben et al., 2020; Kollat et al., 2012; Lane et al., 2019) are now all included in the reference list.

*RC: "I believe an additional discussion point should be added regarding the choice of a particular hydrological model. It is well known that some models may be more flexible than others at adapting to biases in input variables such as precipitation and easily scale PET with specific calibration parameters. This is mentioned in the paper, but I believe some other hydrological models may perform better than the one used in this study, and the performance drop mentioned in this study may not be as bad. I certainly would not expect the conclusions of this paper to be any different, but this should be mentioned."*

We added text to section 4.3 (lines 425-430 in the revised manuscript) that it is possible that the effect of the Caravan forcing data is different (either smaller or larger) if a different model is used as the sensitivity to the input data may be model dependent. We also do not think that the results will be very different for a different lumped conceptual model but agree that it would be interesting to look deeper into the various effects of the choice of the input data for different models.

*RC: "(Section 4.3) As the authors discuss in this section, hydrological models can generally cope well with poor PET values given that they scale this input variable anyway. What would be nice to add to the discussion is the potential problem of biased parameters. Depending on the model structure, one or more parameters will absorb the bias in the forcing data. This is problematic if the resulting values are used to characterize the system (e.g. Bouaziz et al., 2022, HESS, https://doi.org/10.5194/hess-26-1295-2022 and references therein). Are there parameters in HBV that would show this bias? I could not find a good example in the literature, but it would be interesting to see how stepwise increases in PET are reflected in stepwise bias in a parameter."*

We agree that a further investigation of the effects of the overestimated potential evapotranspiration on the parameters of the HBV model is interesting and that more insight into such effects strengthens the point that a bias in the potential evapotranspiration data is problematic, even if their impact on model performance is limited. We compared the calibrated parameters for all study catchments resulting from scenario I (all Camels forcing data) and scenario V (potential evapotranspiration from Caravan, temperature and precipitation from Camels), i.e., for the datasets that differ only in the potential evapotranspiration forcing (regional/national vs. ERA5-Land estimates). For each catchment, we conducted a Wilcoxon test for each parameter to see if the 100 calibrated parameters were significantly different from each other. For the parameters of the soil routine, the resulting p-value was smaller than 0.001 for the vast majority of the 1252 catchments (90 % for parameters *LP* and *FC* and 95 % for parameter *BETA*). For the remaining eight parameters, the percentage of catchments for which the p-value was smaller than 0.001 varied between 32 % (parameter *K1*) and 61 % (parameter *SFCF*). The percentage of catchments for which the p-value was less than 0.05 varied between 50 % (parameter *K1*) and 97 % (parameter *BETA*). Thus, we concluded that the bias in the potential evapotranspiration data is mainly compensated for by the parameters in the soil routine of the HBV model, as it could be expected (Fig. 1).

[Figure]

*Figure 1: Share of catchments for which the 100 calibrated parameters for scenario I were significantly different (two-sided Wilcoxon test) from the 100 calibrated parameters for scenario V. The different colors indicate the different routines of the HBV model.*

We furthermore investigated how the calibrated parameter values change due to a stepwise increase in the overestimation of the potential evapotranspiration data:

a) We artificially increased the potential evapotranspiration data (originating from the Camels datasets) with the factors 1.5, 2, 3, 4, 5, and 6.
b) We calibrated the model 100 times for each catchment (with the same settings as for all calibrations in the study) for each of these potential evapotranspiration time series. For temperature and precipitation, we used the Camels input data.
c) For each parameter and each factor, we compared the median of the 100 parameter values resulting from the calibration with the biased potential evapotranspiration data with the median of the 100 parameter values from the calibration with the Camels potential evapotranspiration data (i.e., scenario I, considered to be unbiased).

As expected from the results of the earlier analysis (see Fig. 1), the increase in the potential evapotranspiration mainly affected the median parameter values for the parameters in the soil routine *FC*, *LP* and *BETA* (Fig. 2). However, there was also a stepwise change in the calibrated values for the parameters of the snow routine, especially for the parameters *SFCF* and *CFMAX*. These changes indicate that the biased input data impact the whole model system and not only the parameters directly affected by them. This could be expected due to model parameter equifinality.

The stepwise decrease in the calibrated value of the soil parameters *FC* and *BETA* and the stepwise increase of the soil parameter *LP* due to the stepwise increase in the potential evapotranspiration data can be directly explained by the characteristics of the soil routine of the HBV model which controls the amount of water that is evaporated. Over a longer simulation period, the following can be observed:

- A lower value of *FC* (maximum soil moisture storage) means that less water is available in the soil box and thus, less water will be evaporated. In other words, it limits the overall evapotranspiration.
- A higher value of *LP* (soil moisture value, i.e., percentage of *FC* above which the actual evapotranspiration equals the potential evapotranspiration) reduces the overall actual evapotranspiration.
- A lower value of *BETA* (parameter that determines the share of the precipitation and snowmelt that becomes soil water and groundwater) leads to more groundwater recharge, i.e., less water

goes to the soil box from where it could leave the catchment via evapotranspiration, i.e., the overall evapotranspiration is reduced.

These changes in the model parameter values are the reason why the actual evapotranspiration is simulated similarly, regardless of which potential evapotranspiration data are used (i.e., it was similar for the model calibrated with the Caravan potential evapotranspiration data that is too high as for the model calibrated with the the realistic Camels potential evapotranspiration data; see the corresponding comment in section 4.3).

We think that including all these findings in the current manuscript would distract from the main findings, thus we included some text in in section 4.3 to describe that the parameters of the soil routine were affected by the overestimated potential evapotranspiration data and that a bias in evapotranspiration data affects how the catchment and hydrological processes are represented. This is problematic when the parameter values are used to characterize a catchment. Thank you for the recommending the paper by Bouaziz et al. (2022), we included it as an example. More specifically, we changed and added some text in the discussion, namely on lines 453-459 in the revised manuscript. We furthermore added the reference to the paper by Bouaziz et al. (2022) in the reference list.

[Figure]

Figure 2: Change in median parameter values when increasing the potential evapotranspiration ($E_{pot}$) data by a factor (1.5 to 6) compared to the median parameter values for the unbiased potential evapotranspiration data from the Camels datasets. Positive values indicate an increase of the parameter value due to the biased potential evapotranspiration data, negative values a decrease. Note that the y-axes differ for each parameter.

*RC: "In addition to the specific comments regarding the Caravan dataset, are there more general lessons to be learned? E.g. regarding how to benchmark new datasets? This general problem might come up more often in the future in various datasets."*

We think that there are two general lessons that can be learned from this study:

1. Large-sample datasets that will be published in the future will have advantages and disadvantages compared to the existing datasets. This is fine because they will serve different purposes. However, clear statements about the advantages as well as the drawbacks of a dataset or its parts are needed. This requires a clear discussion or even a disclaimer section in the accompanying paper and in a corresponding file in the dataset itself. We think that this should become the standard in the large-sample community to ensure that people use the right datasets for the right purpose.

2. Hydrological models are often used to test for the reliability of meteorological forcing data. Our study clearly indicates that this works fine for precipitation data, but that the approach is much less sensitive to the potential evapotranspiration data. Thus, for the validation of potential evapotranspiration data (and the indices calculated based on these data), other approaches such as simple plausibility tests may be more helpful than testing the reliability of the data with a hydrological model that can compensate for the low-quality data.

We included the first point as two paragraphs spanning lines 510-517 in the revised manuscript. We added a paragraph spanning lines 467-470 in the revised manuscript to address the second point.

*RC: "There should be a mention of the upcoming ERA6 reanalysis. The ERA5 reanalysis used in Caravan will soon be a thing of the past. In addition to improved resolution, ERA6 will have a full overhaul of the model physics, including radiation, which is overestimated in ERA5 and likely part of the PET problem, in addition to the issue discussed above. Based on past history, we can expect a significant performance increase with ERA6. This should be mentioned in the paper. I believe that reanalysis is indeed the future of large-sample hydrology and that merging reanalysis with Deep Learning approaches will produce very high-quality global datasets much sooner than most people think. Already, the merging of deep-learning methods with weather forecasting models promises to revolutionize weather forecasting—exciting times."*

Thank you for pointing this out. We included this point in the first general finding mentioned above, namely we added it on lines 511-513 in the revised manuscript. While we could not find any papers about ERA6 that are available already, we agree that it is important to show that comparisons of reanalysis data and other data may lead to other results in the future.

*RC: "I would suggest the use of PET instead of $E_{pot}$, with the former being a lot more common, in my opinion."*

Even though we see the wide use of PET, we prefer to use (and keep) $E_{pot}$ in this manuscript. As we are already using a lot of abbreviations (all consisting of several capital letters) for the datasets, we think that it is better to use the variant with a subscript for the potential evapotranspiration to make it more distinct. Furthermore, $E_{pot}$ can be used in the formula for the aridity index, whereas the use of PET in an equation would be mathematically incorrect (as it would equal P times E times T). We, thus, did not make any changes in response to this comment.